# The p53 Pathway and Metabolism: The Tree That Hides the Forest

**DOI:** 10.3390/cancers13010133

**Published:** 2021-01-04

**Authors:** Airelle Lahalle, Matthieu Lacroix, Carlo De Blasio, Madi Y. Cissé, Laetitia K. Linares, Laurent Le Cam

**Affiliations:** 1Université de Montpellier, F-34090 Montpellier, France; airelle.lahalle@inserm.fr (A.L.); matthieu.lacroix@inserm.fr (M.L.); carlo.de-blasio@inserm.fr (C.D.B.); laetitia.linares@inserm.fr (L.K.L.); 2IRCM, Institut de Recherche en Cancérologie de Montpellier, F-34298 Montpellier, France; 3ICM, Institut Régional du Cancer de Montpellier, F-34298 Montpellier, France; 4INSERM, Institut National de la Santé et de la Recherche Médicale, U1194, F-24298 Montpellier, France; 5Equipe Labellisée Ligue Contre le Cancer, F-75013 Paris, France; 6Department of Molecular Metabolism, Harvard, T.H Chan School of Public Health, Boston, MA 02115, USA; mcisse@hsph.harvard.edu

**Keywords:** p53 pathway, metabolism, network, cancer, aging, metabolic disease

## Abstract

**Simple Summary:**

The p53 pathway is a major tumor suppressor pathway that prevents the propagation of abnormal cells by regulating DNA repair, cell cycle progression, cell death, or senescence. The multiple cellular processes regulated by p53 were more recently extended to the control of metabolism, and many studies support the notion that perturbations of p53-associated metabolic activities are linked to cancer development. Converging lines of evidence support the notion that, in addition to p53, other key components of this molecular cascade are also important regulators of metabolism. Here, we illustrate the underestimated complexity of the metabolic network controlled by the p53 pathway and show how its perturbation contributes to human diseases including cancer, aging, and metabolic diseases.

**Abstract:**

The p53 pathway is functionally inactivated in most, if not all, human cancers. The p53 protein is a central effector of numerous stress-related molecular cascades. p53 controls a safeguard mechanism that prevents accumulation of abnormal cells and their transformation by regulating DNA repair, cell cycle progression, cell death, or senescence. The multiple cellular processes regulated by p53 were more recently extended to the control of metabolism and many studies support the notion that perturbations of p53-associated metabolic activities are linked to cancer development, as well as to other pathophysiological conditions including aging, type II diabetes, and liver disease. Although much less documented than p53 metabolic activities, converging lines of evidence indicate that other key components of this tumor suppressor pathway are also involved in cellular metabolism through p53-dependent as well as p53-independent mechanisms. Thus, at least from a metabolic standpoint, the p53 pathway must be considered as a non-linear pathway, but the complex metabolic network controlled by these p53 regulators and the mechanisms by which their activities are coordinated with p53 metabolic functions remain poorly understood. In this review, we highlight some of the metabolic pathways controlled by several central components of the p53 pathway and their role in tissue homeostasis, metabolic diseases, and cancer.

## 1. Introduction

Functional inactivation of the p53 pathway is considered a prerequisite to cell transformation. Many studies have shown that somatic or germline mutations of *TP53*, the gene encoding the p53 tumor suppressor, as well as other genetic or epigenetic alterations perturbing the activity of upstream regulators of p53, promote carcinogenesis. Beside its well-described canonical functions involved in DNA repair, cell division, cell death, and cellular senescence, the importance of p53 metabolic activities in cancer progression is gaining momentum. Several recent reviews have highlighted the diversity of the mechanisms by which wild type (WT), but also the mutated forms of p53, are implicated in several major metabolic pathways including glycolysis, oxidative phosphorylation (Oxphos), the pentose phosphate pathway, redox homeostasis, polyamine biosynthesis, as well as amino-acid, nucleotide, and lipid metabolism [1,2,3,4,5].

So far, most studies have focused on the importance of the p53 protein in metabolism, but converging evidence indicates that other key components of this pathway play pivotal functions in metabolism. In this review, we focus on one well-characterized branch of the p53 pathway involving several bona fide oncogenes and tumor suppressors, including the B lymphoma Mo-MLV insertion region 1 homolog (BMI1) protein, the Alternative Reading Frame (ARF) tumor suppressor, the Mouse Double Minute 2 (MDM2) oncogene and its partner MDM4, as well as the multifunctional E4F1 protein and p53. Extensive characterization of this molecular cascade showed that BMI1 is an important component of the polycomb repressive complex 1 (PRC1) which controls at the epigenetic level *Cdkn2a*, the locus encoding ARF, a potent inhibitor of the Mouse Double Minute 2 (MDM2) protein that negatively regulates p53. The MDM4 oncoprotein and the E4F1 protein also play important regulatory functions in this pathway. MDM2 and MDM4 heterodimerize through their respective RING domains to form a ubiquitin E3 ligase complex that potentiates p53 proteasomal-mediated degradation. E4F1 was initially identified as a transcription factor targeted by the viral oncoprotein E1A, but was later described as an atypical ubiquitin E3 ligase that controls p53 transcriptional functions independently of protein stability [6,7]. E4F1, ARF, and p53 can form a ternary complex involved in cell cycle regulation, an activity which also reflects E4F1 ability to regulate the CHK1-dependent DNA-damage checkpoint [8,9,10]. E4F1 directly binds to BMI1, and both proteins contribute to hematopoietic and epidermal stem cell function [11,12]. Deregulation of this BMI1-ARF-MDM2/MDM4-E4F1-p53 pathway has been widely associated to oncogenesis. In addition, it also plays a major role in cellular senescence, an activity through which it may influence aging [13]. An important mechanism by which these proteins influence cancer progression and aging involves the control of p53-associated metabolic functions. Nevertheless, several studies indicate that these multifaceted proteins exert multiple metabolic activities independently of p53, highlighting the underestimated complexity of the metabolic network regulated by the p53 pathway (Figure 1). It is noteworthy that many of these proteins shuttle between different subcellar compartments including the cytosol, the nucleus, the nucleolus, and mitochondria where they exert distinct metabolic functions. The tight regulation of their subcellular localization contributes to the control of metabolic enzymes or regulators, but also to the transcriptional and post-transcriptional control of metabolic genes encoded by the nuclear or the mitochondrial genomes (Figure 2). In this review, we summarize our current understanding of the diverse metabolic functions played by these central components of the p53 pathway and how their perturbation contributes to metabolic diseases and cancer progression.

## 2. The p53 Pathway Controls Multiple Metabolic Pathways

Several components of the p53 pathway control the production/utilization of several classes of metabolites including amino-acids, lipids, and nucleotides. Although we still lack an exhaustive vision of the complex metabolic network controlled by the p53 pathway, growing evidence indicates that it contributes to many adaptive responses to changes in nutrients and oxygen availability (Figure 3).

### 2.1. Modulation of Mitochondrial Functions by the p53 Pathway

Mitochondria, the power factory of the cell, represents a metabolic hub that integrates multiple signals originating from the p53 pathway. The p53 protein controls many mitochondrial functions, including replication and integrity of the mitochondrial genome, mitochondrial architecture and dynamics, activity of the electron transport chain (ETC), as well as the flux of several metabolic pathways that take place in mitochondria. Different laboratories have confirmed the localization of p53 in mitochondria where it directly controls cell death as well as mitochondrial respiration independently of its transcriptional functions [2,3]. More recently, mitochondrial pools of MDM2, MDM4, ARF, and BMI1 have also been described. Mitochondrial MDM2 controls the activity of the ETC and regulates mitochondria network dynamics independently of p53. Thus, in response to oxidative stress or hypoxia, MDM2 translocates to the mitochondrial matrix where it preferentially binds to the Light Strand Promoter (LSP), leading to transcriptional repression of *NADH-Dehydrogenase 6* (*MT-ND6*), a gene of the mitochondrial genome encoding an important complex I subunit of the ETC. This mitochondrial function of MDM2 occurs both in p53-proficient and in p53-deficient cells, and does not involve its ubiquitin E3 ligase activity [14]. The role of MDM2 in regulating mitochondrial respiration extends to cytosolic MDM2, which binds and sequesters in the cytosol NADH:ubiquinone oxidoreductase 75 kDa Fe-S protein 1 (NDUFS1), a large Complex I subunit that promotes oxidative phosphorylation and favors super complex formation [15]. MDM2 plays an important role both upstream and downstream of the ETC. Indeed, a screen aiming at identifying synthetic lethal approaches based on drugs inhibiting metabolism highlighted an unsuspected role for MDM2 in sensing impaired bioenergetics triggered by pharmacological inhibition of complex I in conditions of high alpha-ketoglutarate (αKG) levels. Such a combination led MDM2 to regulate alternative exon usage affecting genes involved in glycolysis, thereby resulting in the complete inhibition of glycolysis and the induction of a lethal energetic crisis [16]. Other aspects of mitochondrial homeostasis are controlled by MDM2. When mitochondrial membrane potential is altered and respiration is not efficiently conducted, the recycling of damaged mitochondria is necessary. This physiological process, called mitophagy, is controlled, at least in part, by PARKIN, a ubiquitin E3 ligase that ubiquitinates MITOFUSIN1 on the mitochondrial outer membrane. The direct interaction between MDM2 and PARKIN enhances PARKIN enzymatic activity (self-ubiquitination and MITOFUSIN1 ubiquitination), thereby promoting mitophagy [17]. Interestingly, the links between the p53 pathway and PARKIN extend to p53-mediated control of *PARKIN* transcription and to p53-PARKIN protein-protein interaction [18,19,20,21]. Finally, p53 also controls the transcription of *SPATA18* (also called *MIEAP*), which gene product is involved in intramitochondrial lysosome-like structures that eliminate oxidized mitochondrial proteins and thereby improve mitochondrial functions [22]. These data reinforce the notion that the p53 network plays a central role in quality control and mitochondria turnover.

Other key regulators of the p53 pathway control mitochondrial functions beyond their effects on p53. BMI1 represses the transcription of genes (*Alox5*, *Alox15*, *Cyp24a1*, *Cyp26a1*, *Bnip3l*, *Pmaip1*, *Duox1*, *Duox2*, *Cdo1*) encoded by the nuclear genome that influence mitochondrial function and redox homeostasis independently of its role on the epigenetic control of the *Cdkn2a* locus. Consistently, cells isolated from *Bmi1* knock-out (KO) mice display impaired mitochondrial respiration and a marked increase in the intracellular levels of reactive oxygen species (ROS) that lead to the engagement of the CHK2-dependent DNA damage checkpoint [23]. BMI1 activity in metabolism also involves its mitochondrial localization where it directly controls polynucleotide phosphorylase, a ribonuclease responsible for mitochondrial RNA (mtRNA) transcripts decay, thereby regulating mtRNA homeostasis and bioenergetics [24]. In addition, the multifunctional protein E4F1, through its intrinsic transcriptional activity, and independently of its actions on p53, controls the expression of genes encoding a complex I subunit (*Ndufs5*) or a component of the mitochondrial import machinery (*Tomm7*), as well as gene sets involved in cardiolipin (a mitochondria-specific phospholipid) biosynthesis and maturation (*Dnajc19*, *Crls1*, *Taz*) or in pyruvate oxidation (*Mpc1*, *Dlat*, *Dld*, *Pdpr*, *Slc25A9*). Consistently, genetic inactivation of *E4f1* in murine transformed fibroblasts impacts on oxygen consumption and other metabolic pathways that are compartmentalized in mitochondria (see below), confirming that this E4F1-controlled transcriptional program strongly influences cellular metabolism [9]. Finally, the MDM4 oncoprotein and the N-terminally truncated isoform of the ARF tumor suppressor that is generated upon translation initiation at an internal in-frame AUG codon at position 45 (also called small mitochondrial Arf or p15smArf), have both been detected in mitochondria [25,26,27,28]. Although these mitochondrial pools of MDM4 and ARF were initially associated to the control of cell death, it is tempting to speculate that, similarly to other components of the p53 pathway exhibiting mitochondrial localization, they also contribute to various mitochondrial activities involved in metabolism. Altogether, these studies indicate that the p53 pathway is tightly connected to mitochondria functions to fine tune metabolism.

### 2.2. Implication of the p53 Pathway in Pyruvate Metabolism

Pyruvate is a central metabolite that stands at the crossroads of glycolysis and Oxphos. Pyruvate is imported in mitochondria to fuel the tri-carboxylic acid (TCA) cycle and sustain mitochondrial respiration, but also to contribute to several anabolic pathways implicated in de novo fatty acid and cholesterol synthesis, gluconeogenesis, and nucleotide metabolism. One key enzyme of pyruvate metabolism is the pyruvate dehydrogenase (PDH) complex (PDC), a large multi-subunit metabolic enzyme that converts pyruvate into Acetyl-CoenzymeA (AcCoA) in the mitochondrial matrix. Decreased PDC activity redirects pyruvate metabolism towards lactate or alanine production by the lactate dehydrogenases (LDH) and alanine amino-transferases (ALAT), respectively. The p53 pathway is linked to the PDC at multiple levels. Thus, whereas p53 was found to repress the transcription of *PDK2*, a gene encoding an inhibitory kinase of the PDC, MDM2 inhibition stabilizes the protein levels of dihydrolipoamide dehydrogenase (DLD), the E3 subunit of the PDC [29,30]. Furthermore, the shuttling of the MDM2-DLD complex between the cytosol and the nucleus is modulated by the pharmacological inhibitor Nutlin3A, which interferes with p53-MDM2 interaction. Strikingly, the E4F1 protein was also identified as a key regulator of the PDC. Thus, E4F1 directly controls the transcription of genes encoding several essential subunits and regulators of the PDC in various cell types, including the E2 and E3 subunits of the PDC core enzyme (*Dlat*, *Dld*), the mitochondrial pyruvate carrier (*Mpc1/Brp44l*), the mitochondrial transporter of the PDH co-factor Thiamine Pyrophosphate/TPP (*Slc25a19*), and the regulatory subunit of the PDC phosphatases (*Pdpr*). Consistent with its role in the regulation of the PDC, stable isotope tracing experiments using ^13^C-labelled glucose showed that *E4f1* inactivation, decreases glucose-derived AcCoA production and enhances lactate production [31,32,33]. Perturbation of E4F1 functions in pyruvate metabolism has clinical implications. Indeed, a non-synonymous homozygous mutation (K144Q) in the coding region of the human *E4F1* gene was recently identified in two siblings of an Italian family presenting clinical symptoms reminiscent of those of Leigh syndrome patients [34]. The Leigh syndrome is a severe inborn metabolic disorder characterized by a progressive subacute necrotizing encephalomyelopathy resulting from various mitochondrial defects affecting the PDC or the ETC. Consistent with these findings, genetically engineered mouse models lacking E4F1 in their skeletal muscles or in the central nervous system display phenotypes that recapitulate some of the clinical symptoms of Leigh syndrome patients, including chronic lactate acidemia, muscular endurance defects, microcephaly, and neuronal degeneration [33]. Finally, the tight connections between the p53 pathway and pyruvate metabolism are also illustrated by findings showing that MDM2 and p53 are part of a regulatory network in pancreatic beta cells controlling the activity of pyruvate carboxylase (PC), thereby influencing glucose-stimulated insulin secretion and glucose homeostasis [35]. Altogether, these data indicate that the p53 pathway is closely linked to pyruvate metabolism in many cell types and that perturbations of this complex network contribute to various human diseases, including inborn metabolic disorders, type-II diabetes, and cancer.

### 2.3. Role of the p53 Pathway in Amino-Acid Metabolism

p53 has been linked to several aspects of glutamine, serine/glycine, and proline metabolism, and p53-deficient cells are more sensitive to serine/glycine or to glutamine deprivation [36,37,38,39,40,41,42,43]. The control of MDM2 subcellular localization is also implicated in amino-acid metabolism, as illustrated by findings showing that MDM2 is recruited to chromatin independently of p53 to regulate genes involved in serine/glycine, as well as in glutamine/glutamate, metabolism. Through its binding to the ATF4 transcription factor, chromatin-bound MDM2 activates a transcriptional program composed of several genes encoding transporters involved in serine uptake (SLC1A4) or its intracellular processing (SERINC1), as well as enzymes implicated in de novo serine synthesis, an anabolic pathway that converts the glycolytic intermediate 3-phosphoglycerate (3PG) into serine through a multi-step enzymatic cascade implicating phosphoglycerate dehydrogenase (PHGDH), phosphoserine aminotransferase 1 (PSAT1), and phosphoserine phosphatase (PSPH). The recruitment of MDM2 to chromatin is triggered by serine and glycine deprivation, oxidative stress, or upon inhibition of the M2 isoform of pyruvate kinase (PKM2), a glycolytic enzyme for which serine is an allosteric activator [44]. Recruitment of MDM2 to chromatin, which likely involves conformational changes implicating its central acidic domain, but occurs independently of its E3 ligase function, is inhibited by its phosphorylation on serine 166 and threonine 351, an event modulated by the glycolytic enzyme PKM2 [45]. Strikingly, chromatin-bound MDM2 and p53 display antagonistic activities on the transcription of genes involved in serine metabolism. Thus, whereas MDM2 activates the transcription of *SLC1A4*, *SERINC1*, *PHGDH*, *PSAT1*, and *PSPH*, p53 was previously found to repress the *PHGDH* promoter, illustrating the complex interplay between the p53 pathway and serine metabolism [45,46]. Serine/glycine and glutamine/glutamate metabolism contribute to various anabolic pathways, including glutathione (GSH) and nucleotide biosynthesis [47]. Consistent with its role in these metabolic pathways, chromatin-bound MDM2 was shown to influence the redox status of both normal and cancer cells through the regulation of glutathione synthesis and recycling [45]. The notion that MDM2 is a central player of serine metabolism was recently confirmed in liposarcomas (LPS), a sarcoma subtype characterized by a systematic amplification of *MDM2*. Interestingly, LPS, but not other sarcoma subtypes, display high levels of chromatin-bound MDM2. These data support the notion that the strong selective pressure for *MDM2* amplification in LPS likely reflects its predominant role in serine metabolism to support nucleotide synthesis in these highly proliferating cancer cells. Interestingly, this study pointed at one important limitation of therapeutic strategies based on the utilization of Nutlin3A, a well-characterized compound targeting MDM2-p53 interaction. Thus, Nutlin3A stabilized p53, but unexpectedly promoted MDM2 recruitment to chromatin and the activation of its metabolic target genes in LPS cells, providing a molecular explanation for the poor clinical efficacy of this class of MDM2 inhibitors in LPS patients. In contrast, genetic or pharmacological inhibition of chromatin-bound MDM2 by SP141, a distinct MDM2 inhibitor that triggers its degradation, or interfering with serine metabolism, efficiently impaired LPS growth in pre-clinical models, providing a strong rationale for new therapeutic approaches based on drugs targeting MDM2-mediated control of serine metabolism in LPS [48].

### 2.4. The p53 Pathway and Nucleotide Metabolism

In LPS cells, chromatin-bound MDM2 promotes both purine and pyrimidine biosynthesis independently of p53 [48]. Paradoxically, cytosolic MDM2 has been shown to monoubiquitinate and reduce the activity of Dihydrofolate Reductase (DHFR), a key enzyme involved in folate metabolism that generates precursors for purine synthesis [49]. Interestingly, MDM2 localization in the nucleolus was previously shown to be regulated by the binding of adenine-containing nucleotides to the Walker A or P loop motif and conformational changes of its C terminus domain [50]. Altogether, these data illustrate the importance of the different cellular pools of MDM2 in nucleotide metabolism. In addition to being crucial for nucleotide synthesis, MDM2 is also involved in the repair of oxidized bases through its ubiquitin E3 ligase function. Indeed, during oxidative stress, it contributes to the opening of chromatin to promote Base Excision Repair (BER) via Histone 2B ubiquitination, a process involving its phosphorylation by the kinase MPS1 [51]. Finally, inhibition of the multifunctional protein E4F1 in p53-deficient cells, through a yet unidentified molecular mechanism impinging on orotate metabolism, profoundly affects pyrimidine, but not purine, biosynthesis [9]. Although these data illustrate various mechanisms by which MDM2 and E4F1 contribute to nucleotide metabolism independently of p53, it is likely that these activities are somehow coordinated with p53-mediated control of nucleotide synthesis which occurs, at least in part, through the regulation of the *Ribonucleotide Reductase* (*RRM2*) gene [52,53]. Proper coordination of this complex metabolic network likely contributes to sustain the strong demand in nucleotides of rapidly proliferating cells.

### 2.5. The p53 Pathway, Lipid Metabolism and Adipocyte Cell Fate

An important aspect of p53 metabolic activities relates to its multiple roles in lipid transport and storage, in fatty acids biosynthesis and their desaturation, in cholesterol and sphingolipid metabolism, as well as in fatty-acid oxidation (FAO). Moreover, several groups have documented that p53 interferes with adipocyte differentiation [3,54]. Interestingly, MDM2, MDM4, and BMI1 are also involved in lipid metabolism and adipocyte differentiation, supporting the notion that the entire p53 pathway is pivotal for lipid homeostasis in stressed and cancer cells, as well as in normal tissues. The importance of p53-mediated control of lipid metabolism was nicely illustrated using genetically engineered mouse models exhibiting impaired MDM2 or MDM4 activity. Indeed, mice harboring the *Mdm2*^C305F^ allele, which encodes a mutant exhibiting impaired binding to the RPL11 and RPL5 ribosomal proteins, display defective response to nutritional challenge due to their inability to control p53-mediated regulation of FAO [55]. Another example of the links between the p53 pathway and fatty acid synthesis comes from the analysis of *Mdm4* KO mice expressing a p53 acetylation mutant harboring lysine to arginine (K to R) substitutions on the key lysines 117, 161, and 162 of its DNA binding domain (also called p53^3KR^) that is unable to induce cell cycle arrest, apoptosis, and senescence, but remains competent for its metabolic activities [56,57]. Strikingly, these *Mdm4*^KO^; *p53*^3KR^ compound mice are resistant to high-fat diet (HFD)-induced obesity, a phenotype that was attributed to enhanced FAO. At the molecular level, this effect was linked to the transcriptional regulation of *Long-chain fatty acid elongase 3* (*Elovl3)*, a p53-target gene influencing adipocyte cell fate and energy expenditure. Importantly, MDM2 and BMI1 were also suggested to control adipocyte cell fate determination independently of p53. This MDM2 activity was linked to its ability to regulate cAMP-mediated induction of CCAAT/enhancer-binding protein delta (C/EBPΔ) expression by facilitating the recruitment of the cAMP regulatory element-binding protein (CREB)-regulated transcription coactivator (CRTC2) to the *c/EBPΔ* promoter [58]. Finally, the polycomb member BMI1 suppresses adipogenesis of bone marrow stromal progenitors in the hematopoietic stem cell niche through the epigenetic control of a PAX3-regulated developmental program, explaining some of the *Cdkn2a*-independent cell-extrinsic effects of BMI1 deficiency on hematopoietic stem cell maintenance [59]. Hence, these results indicate that the p53 pathway exerts multiple functions influencing adipocyte cell fate and lipid homeostasis, and this has a major impact on tissue homeostasis, metabolic diseases, aging, and cancer development.

### 2.6. The p53 Pathway and Iron Metabolism

Controlling iron levels is vital for cell survival, and iron overload promotes carcinogenesis. p53 and MDM2 are pivotal in a complex network influencing iron uptake, storage, and usage, both at the systemic and the cellular levels [60]. p53 regulates the transcription of several key iron regulators including *Hepcidin* (*HAMP*), *iron-sulfur cluster assembly enzyme* (*ISCU*), *Ferredoxin reductase* (*FDXR*), and *Frataxin* (*FXN*) [61,62,63,64,65]. Moreover, its activities are directly modulated by intracellular iron levels [66,67]. Changes in free iron levels also modulate *MDM2* mRNA and protein levels, at least in part through the binding of IRP2 to the 3′ untranslated region (3′UTR) of *MDM2* mRNA, thereby influencing indirectly p53 protein stability [68,69]. The links between high iron levels and enhanced MDM2-mediated degradation of p53 in hepatocytes may contribute to the increased risk of hepatocellular carcinoma in patients affected by chronic liver disease [68].

One important biological process associated to the control of iron metabolism by the p53 pathway is ferroptosis, an iron-dependent cell death mechanism linked to lipid peroxidation [4,70]. Beside the well-recognized role of p53 in ferroptosis, which involves the transcriptional and the non-transcriptional regulation of inducers and regulators of ferroptosis, recent work from Venkatesh et al. showed that MDM2 and MDM4 interfere with the ability of cells to build up defenses against lipid peroxidation. Inhibition of MDM2 and/or MDM4 allows cells to accumulate endogenous lipophilic antioxidants such as CoenzymeQ10 (CoQ), an effect mediated by PPARα and FSP1 [71]. Interestingly, p53-mediated control of the mevalonate pathway has also been shown to contribute to CoQ biosynthesis, suggesting that p53, MDM2, and MDM4 control synergistic metabolic functions converging on this key metabolite [72,73].

### 2.7. The p53 Pathway Is Highly Connected to Autophagy

Previous studies underlined the importance of p53 in the autophagy network to promote cell survival. However, p53 can also trigger autophagic cell death in various severe stress conditions [74]. p53 can regulate mitophagy as well as macroautophagy, a process leading to the synthesis of double-membrane vesicles and their fusion to lysosomes to recycle macromolecules and maintain intracellular pools of metabolites. Depending on its subcellular localization, its mutational status, and stress type, p53 can inhibit or stimulate autophagy through multiple mechanisms including the transcriptional control of many autophagy-related genes, the regulation of the mTORC1 kinase which tightly controls the autophagic process according to the intracellular energy and nutrient levels, through the regulation of BLC2 family members which also control autophagy, or upon direct interaction with the key autophagic regulator BECLIN 1 [75,76,77,78,79,80]. Several studies suggest that the links between the p53 pathway and autophagy extend to other components of this cascade, including E4F1 and MDM2. Thus, genetic inactivation of *E4f1* has been shown to induce autophagy in leukemic cells [81] and MDM2 was also shown to be regulated upon accumulation of the autophagy substrate p62/SQSTM1 in KRAS^G12D^-driven pancreatic cancer cells [82]. In addition, MDM2 controls the balance between apoptosis and autophagy in Nutlin-treated cells. Indeed, Nutlin blocks autophagy and promotes apoptosis in MDM2-amplified cancer cells, whereas it promotes autophagy in MDM2 non-amplified cells. This differential effect was associated with αKG levels and the transcriptional regulation of ATG genes, through an epigenetic mechanism implicating the regulation of the αKG-dependent demethylase JMJD2b [83,84]. Several laboratories have also reported that ARF can modulate autophagy, at least in part through its direct interaction with BCL-XL that negatively regulates BECLIN1. Although conflicting results were reported regarding the roles played by smARF and full-length ARF in autophagy, shRNA-mediated silencing of ARF in a B cell lymphoma model impaired autophagy and tumor growth, suggesting that ARF plays a cytoprotective function in some, but not necessarily all, cancer cells [26,85,86,87]. Finally, the polycomb member BMI1 was shown to directly bind and repress *Cyclin G2* (*CCNG2*) in CML cells to control autophagy by activating the PKC-AMPK-JNK-ERK pathway [88].

### 2.8. The p53 Pathway and Hypoxia

Cells adapt their metabolism to changes in nutrient and oxygen availability. Decreased oxygen concentration triggers a complex cellular response coordinated by the Hypoxia-Inducible-Factor (HIF) transcription factors, as well as different components of the p53 pathway. This biological process includes various metabolic adaptations during which p53, but also MDM2 and MDM4, modulate the activity of the ETC and the production of ROS as well as the levels of anti-oxidant molecules to influence cell survival. Initially, HIF1α, the limiting partner of this heterodimeric transcription factor, was shown to directly bind and stabilize p53 [89]. Since, other mechanisms leading to p53 activation have been proposed, such as HIF1α interaction with MDM2, a process inhibiting MDM2-mediated degradation of p53 and which bridges p53 to HIF1α [90]. The downregulation of MDM2 upon its phosphorylation by the p38 mitogen-activated protein kinase contributes to p53 activation in hypoxic neuronal cells [91]. Furthermore, a proteomic study aiming at characterizing new partners of the von Hippel Lindau (VHL) tumor suppressor, an essential component of the ubiquitin E3-ligase complex that mediates proteasomal-mediated degradation of HIFs, identified ARF as a partner of the long isoform of VHL. ARF disrupts the E3 ligase complex containing VHL and instead enhances its interaction with the arginine methyltransferase PRMT3 which methylates p53 [92]. Several other key components of the p53 pathway can also contribute to the cellular response to hypoxia independently of p53. As previously mentioned, MDM2 translocates to the mitochondrial matrix where it represses the transcription of *MT-ND6* and thereby specifically impacts on complex I activity. Strong evidence associates uncoupling of the ETC to the production of mitochondrial ROS by complex I and/or complex III. Consistent with its role in the regulation of complex I activity in hypoxic cells, enhanced recruitment of MDM2 to mitochondria leads to increased mitochondrial ROS levels. Moreover, mitochondrial MDM2 plays a physiological role in muscular cells in response to low oxygen conditions. Indeed, mice lacking MDM2 and p53 in their skeletal muscles display increased ND6 levels that correlate with higher complex I activity, and consistently, *MDM2*; *p53* double KO animals exhibit increased muscular endurance in mild hypoxic conditions when compared to *p53* KO mice. Interestingly, increased mitochondrial-MDM2 levels enhance the migratory and invasive properties of cancer cells, suggesting that mitochondrial-MDM2 could also increase cancer cell aggressiveness in tumoral hypoxic areas [14]. In contrast, ARF antagonizes hypoxia-induced migration of cancer cells through its direct binding to the COOH-terminal binding protein (CtBP) family of metabolically-regulated transcriptional co-repressors [93]. Finally, MDM4 also plays a significant role in modulating p53 activities in hypoxic conditions, a process involving MDM4 phosphorylation by the CHK1 kinase and its subsequent sequestration in the cytoplasm by the 14.3.3 protein [94]. Hence, these data indicate that many components of the p53 pathway contribute to coordinate the cellular response to hypoxia.

## 3. Discussion

A rapidly increasing number of studies support the notion that many, if not all, components of the p53 pathway are key metabolic regulators and that their metabolic functions extend beyond their ability to control the p53 protein. Through their implication in various metabolic pathways, many of these p53 regulators contribute to the three main outputs of metabolism: bioenergetics, biomass production, and maintenance of redox homeostasis. The data mentioned in this review illustrate interesting connections between several components of the p53 pathway and various metabolic pathways (Figure 4). Their importance in metabolism also involves the regulation of other key biological processes such as autophagy and the control of metabolite uptake and export.

There is currently no unifying model explaining how p53 and its many regulators coordinate metabolism, but this notion raises important questions relevant to many physiological and pathological contexts including aging, metabolic diseases, and cancer. First, if p53 is the major downstream effector of this pathway, one may wonder why these different components of the p53 pathway contribute to metabolism through both p53- dependent and independent mechanisms? In comparison to a linear pathway, a branched network is more adapted to respond to multiple types of metabolic challenges. In normal cells, such adaptive responses would require dynamic changes of small amplitude occurring at multiple levels of these highly plastic metabolic networks but which, altogether, ensure cellular homeostasis. Although speculative at the moment, we propose a model where the coordinated metabolic functions of all these regulators of the p53 pathway play synergistic functions that help cells to cope with the multiple metabolic challenges they face on a daily basis, thereby providing a fitness advantage in the long term. Another possibility is that these various components of the p53 pathway play distinct roles as sensors, mediators, or effectors during these metabolic responses. p53 is a major effector of many metabolic stress responses by controlling the transcription of numerous metabolic genes [3]. Nevertheless, several studies have shown that its DNA binding properties are directly controlled by intracellular ROS, heme, or ceramide levels, supporting the notion that p53 is also a bona fide metabolic sensor [66,95,96]. MDM2 plays a central role as a mediator in the cellular response to changes in serine levels. Indeed, its recruitment to chromatin stands between the metabolic sensor PKM2, a glycolytic enzyme which has its activity modulated by intracellular levels of serine, and the ATF4 transcription factor that controls the expression of genes implicated in de novo serine synthesis and serine transport [45]. It will be interesting to investigate whether the activities of other components of the p53 pathway are directly modulated by metabolites through conformational changes affecting protein–protein interactions, DNA binding, subcellular localization, or E3 ligase function. A non-mutually exclusive explanation is that the various metabolic activities regulated by p53, MDM2, MDM4, BMI1, ARF, and E4F1 define important feed-back loops that guarantee the proper control of cellular responses to changes in nutrient/oxygen concentrations over time. Thus, it is possible that when cells face a transient decrease in intracellular serine levels, they initially activate de novo serine synthesis by chromatin-bound MDM2, but later induce p53-mediated repression of *PHGDH* through a yet-to-be defined mechanism to bring back the activity of this key anabolic pathway to basal levels [45,46]. Many other feed-back loops operating within this molecular cascade are likely to fine tune metabolism according to the levels of several key metabolites. Recent findings showing that MDM2 controls glycolysis independently of p53 upon energetic shortage resulting from ETC-CI deficiency in conditions of high αKG levels, together with observations indicating that restoring WT-p53 functions in pancreatic cancer cells results in accumulation of αKG, suggest that this key metabolite is central to p53-associated metabolic networks [16,97]. Altogether, these converging lines of evidence indicate that the p53 pathway controls a highly ramified metabolic network that is essential to maintain cellular homeostasis.

Second, it is interesting to consider these complex metabolic functions from an evolutionary standpoint. The observation that p53, and perhaps other key regulators of the p53 pathway, initially favor cell survival in conditions of nutrient deprivation, led to the hypothesis that one of their evolutionary conserved functions is to protect, both at the cellular and at the systemic levels, body integrity in conditions of limited access to nutrients. This notion was initially proposed by Murphy and colleagues in studies aiming at understanding functional differences of the arginine and proline variants at codon 72 of the *TP53* gene. Interestingly, population-based studies and genetically-engineered mouse models indicate that this single nucleotide polymorphism (SNP) is not associated to higher cancer incidence, but rather to increased body weight and enhanced risk for diabetes [98]. Compared to cells expressing the proline variant (P72), those harboring the arginine variant (R72) display increased cell survival in conditions of nutrient deprivation, but not upon genotoxic stress [99]. These findings led to the hypothesis that the R72 allele may have been selected in populations in which a better response to nutrient deprivation provides a selective advantage. It is noteworthy that some Li-Fraumeni patients with germline mutations of *TP53* have been shown to display an increased Oxphos capacity in their skeletal muscles [100]. These genetic traits confirm the tight links between p53 and metabolism, a function that may have influenced their selection in human populations. Genetic studies have linked other SNPs in the *MDM2* and *MDM4* genes to increased cancer incidence or premature aging [101,102,103,104,105,106]. Evaluating whether some of these polymorphisms also correlate with metabolic phenotypes may unravel new interesting connections between MDM2/MDM4 and metabolic diseases. Another striking illustration of the importance of these metabolic activities in human diseases is the recent identification of Leigh syndrome patients harboring a homozygous mutation in the *E4F1* gene [34]. It is tempting to speculate that this Italian family harboring the *E4F1*^K144Q^ mutation represents just the tip of the iceberg, and that future studies will link other components of the p53 pathway to inborn metabolic disorders of currently unknown etiology. Although the identification of these families may require a massive sequencing effort, the characterization of their associated clinical symptoms will likely improve our understanding of the complex roles of the p53 pathway in metabolism.

Finally, the complexity of the metabolic network regulated by the p53 pathway has direct clinical implications for the design of new cancer therapies. It is time to reconsider the over-simplistic concept that all the genetic alterations occurring at various levels on the p53 cascade lead to the same consequence, e.g., functional inactivation of p53. At least from a metabolic standpoint, genetic alterations of these different components of the p53 pathway will rewire metabolism very differently. Moreover, the cellular contexts in which they occur will dramatically influence the associated molecular consequences. Although challenging, characterizing in a systematic manner the multiple metabolic defects triggered by the most common genetic alterations of the p53 pathway and integrating them into computational models should provide a more complete picture and help the design of rationalized therapies targeting potential metabolic bottlenecks.

## Figures and Tables

**Figure 1 cancers-13-00133-f001:**
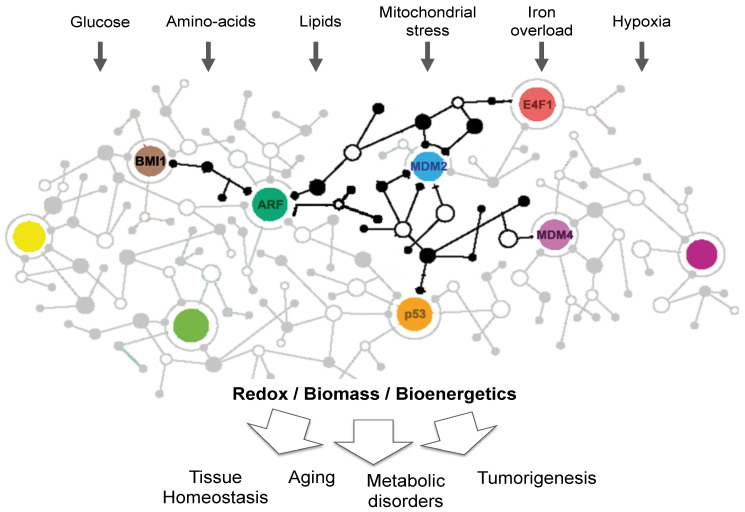
The p53 network and metabolism. Various key components of this molecular cascade sense changes in oxygen or nutrient concentrations and ensure tissue homeostasis by regulating biomass production, bioenergetics, and redox homeostasis through multiple metabolic pathways. Perturbations of this complex network contribute to aging, metabolic disorders, and cancer progression.

**Figure 2 cancers-13-00133-f002:**
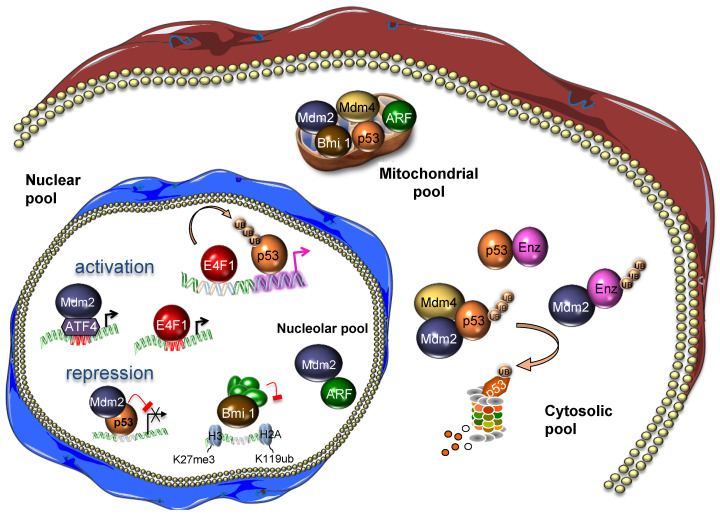
The BMI1, ARF, MDM2, MDM4, E4F1, and p53 proteins shuttle between different subcellular compartments where they control different metabolic functions. Enz, metabolic enzyme; Ub, ubiquitin.

**Figure 3 cancers-13-00133-f003:**
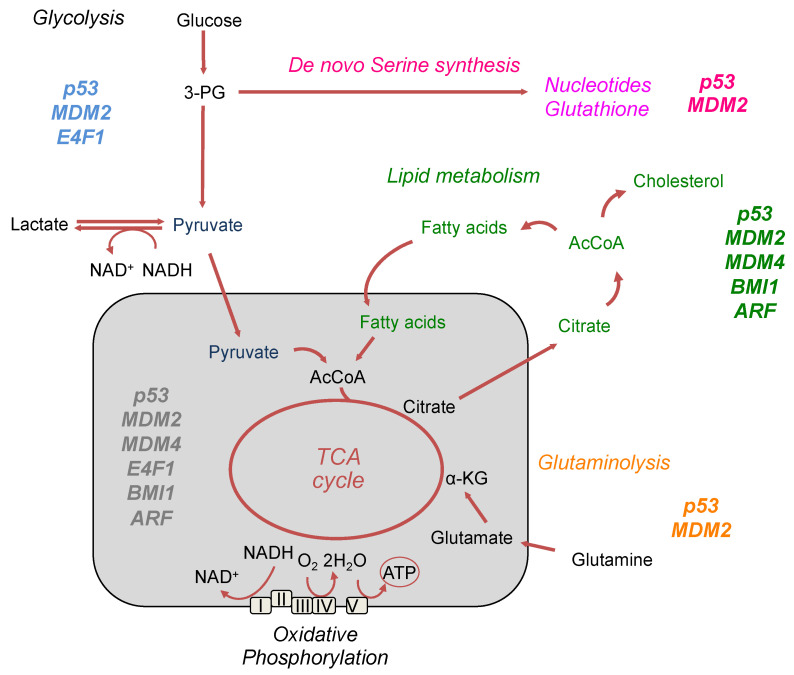
Schematic representation of several metabolic pathways in which the different components of the p53 network are implicated, including glycolysis, Oxphos, amino acid, and fatty acid metabolism.

**Figure 4 cancers-13-00133-f004:**
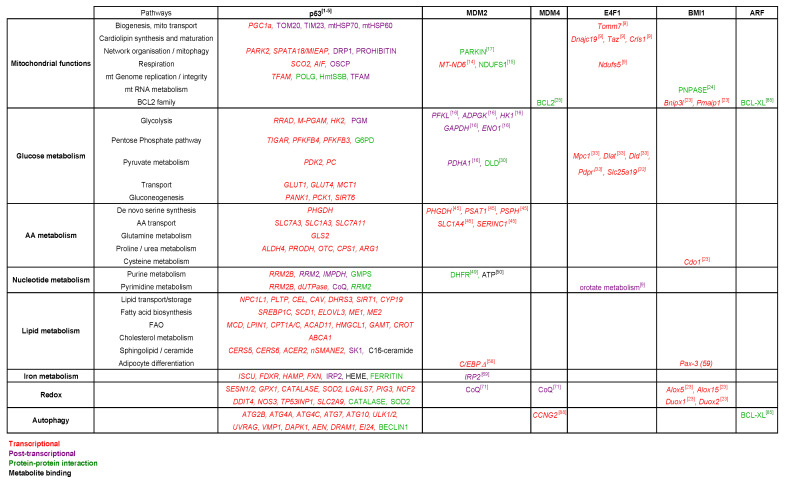
Roles of MDM2, MDM4, ARF, BMI1, and E4F1 in various metabolic pathways.

## Data Availability

Not applicable.

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
