# Peer review of "The p53 Pathway and Metabolism: The Tree That Hides the Forest"

_cancers, 2021, doi:10.3390/cancers13010133_

Round 1

Reviewer 1 Report

In this review, Airelle et al. provide an extensive review about the role of some components of the p53 signalling (p53, BMI1, ARF, E4F1, MDM2, MDM4) in the regulation of several metabolic pathway. Overall, it is an enjoyable reading that highlights the importance of p53 pathway beyond its classical function such as DNA damage response and senescence.

Comments:

  • Line 40: p53 pathway “is” considered, instead of “in” considered
  • Figure 1: while the authors suggest a non-linear p53 pathway in the manuscript, this scheme illustrates a linear pathway. Also, the three protein complexes are not linked to the pathway, but rather they are floating in the middle. I would divide this figure in 2 subsections, (A) illustrating the different protein complexes and (B) the scheme. alternatively, I suggest to link the complexes to the different levels of the scheme, in which they play a role 
  • Figure 2 is missing in the PDF. It is unclear whether Figure 2 in the text (line 93) corresponds to actual figure 3.
  • Table 1 is not readable. I would also add p53 information in this table.
  • Line 452: is it a question or a statement?
  • Title: as the review is not only about p53, but also about its components, I suggest to change the title to p53 pathway or p53 components.
  • figure 4 is very similar to figure 1 and 2, and is not adding extra-information or facilitating the text. 

Author Response

We answered to all the constructive comments of the reviewers.

Please find below our answers to reviewer 1 of our manuscript untitled “The p53 pathway and metabolism: the tree that hides the forest”.

Reviewer 1:

In this review, Airelle et al. provide an extensive review about the role of some components of the p53 signalling (p53, BMI1, ARF, E4F1, MDM2, MDM4) in the regulation of several metabolic pathway. Overall, it is an enjoyable reading that highlights the importance of p53 pathway beyond its classical function such as DNA damage response and senescence.

Comments:

  • Line 40: p53 pathway “is” considered, instead of “in” considered
    The sentence has been changed.

Figure 1: while the authors suggest a non-linear p53 pathway in the manuscript, this scheme illustrates a linear pathway. Also, the three protein complexes are not linked to the pathway, but rather they are floating in the middle. I would divide this figure in 2 subsections, (A) illustrating the different protein complexes and (B) the scheme. alternatively, I suggest to link the complexes to the different levels of the scheme, in which they play a role 
Since several reviewers found that some of the figures were slightly redundant, we removed Fig 1 and replaced it by a modified version of Fig 4. We believe this new Figure 1 illustrates more faithfully one of the main messages of this review regarding the complexity of the metabolic network regulated by the different components of the p53 pathway.

Figure 2 is missing in the PDF. It is unclear whether Figure 2 in the text (line 93) corresponds to actual figure 3.
This was unfortunately due to the software used by the journal which failed to incorporate this figure. This will be fixed during editing of the Ms.

  • Table 1 is not readable. I would also add p53 information in this table.
    The table has been changed to make it easier to read. As requested by one reviewer, we also added information summarizing the main metabolic functions of p53. Note that because of space constraints in this table as well as in the bibliography section, the publications describing the multiple links between the p53 protein and metabolism are not indicated in this table. We refer to several recent good reviews for more details.
  • Line 452: is it a question or a statement?
    This was a typo mistake. We removed the question mark.
  • Title: as the review is not only about p53, but also about its components, I suggest to change the title to p53 pathway or p53 components.
    Thanks for the suggestion. We decided to change the title to “The p53 pathway and metabolism: the tree that hides the forest”.
  • figure 4 is very similar to figure 1 and 2, and is not adding extra-information or facilitating the text. 
    see our previous comment

Reviewer 2 Report

This is a very comprehensive review that would be of interest to both novice and experienced readers in the field. Lahalle et al have done a great job summarizing convoluted roles of p53 pathways in various physiological and pathophysiological conditions. The review, for the most part, reads well, though the language in some sections could be more straightforward. This review should be considered for publication if the minor issues below are corrected.

  1. Figure 2 is missing in my version and couldn't see the figure, not sure if its a software issue.
  2. Table 1 is not very clear either, so could not evaluate it.
  3. In general, the figures could use professional editing.

Author Response

We answered to all the constructive comments of the reviewers.

Please find below our answers to reviewer 2 of our manuscript untitled “The p53 pathway and metabolism: the tree that hides the forest”.

Reviewer 2:

This is a very comprehensive review that would be of interest to both novice and experienced readers in the field. Lahalle et al have done a great job summarizing convoluted roles of p53 pathways in various physiological and pathophysiological conditions. The review, for the most part, reads well, though the language in some sections could be more straightforward. This review should be considered for publication if the minor issues below are corrected.

Figure 2 is missing in my version and couldn't see the figure, not sure if its a software issue.
This was unfortunately due to the software used by the journal which failed to incorporate this figure. This will be fixed during editing of the Ms.

  1. Table 1 is not very clear either, so could not evaluate it.
    The table has been changed to make it easier to read. As requested by one reviewer, it now also summarizes the metabolic functions of p53. Note that because of space constraints in this table as well as in the bibliography section, the publications describing the multiple links between the p53 protein and metabolism are not indicated in this table. We refer to several recent good reviews for more details.

  2. In general, the figures could use professional editing.
    We can’t afford the cost of professional editing but we tried to improve the quality of the figures as much as possible.

Reviewer 3 Report

The review by Lahalle et al. focusses on the metabolic network that is controlled by the p53 pathway. The authors first summarize well studied aspects of p53’s functions in RNA repair, cell division, cell death and aging and then describe the components of a branch within the p53 pathway that pertains to metabolism. Subsequently, they elaborate on eight metabolic pathways controlled by p53 in detail.  

This review is interesting and well written. The figures are informative and I would assume that the proposed links between the p53 pathway and the control of metabolic networks will be inspiring for the field of cancer research.

Minor points:

I think that Figure 1 could be somewhat improved at the visual level (the proteins look like smarties, which I find a little distracting) and senescence should be spelled with a small s. Metabolism is linked to physiological features (muscle endurance), pathologies (obesity, tumorigenesis), a tissue (CNS), a state (stemness) and a biological process (senescence) in a seemingly random order. Could this be structured a bit more?

In my document there is no Figure 2.

Line 134: I suggest using the official gene symbol for Mieap (SPATA18) to avoid confusion.

Line 252: …profoundly affects pyrimidine but not purine biosynthesis.

Line 256: it seems that full gene names are written in italics occasionally (as compared to, for example, line 242). I guess this depends upon the Journal’s style but it should be harmonized throughout the text.

Line 393: should it not read “for which the DNA binding properties are directly controlled by…”?

Line 425: The sentence has a question mark at the end but it is not a question. If the authors intended it to be a question, they should invert “it” and “is”.

Author Response

we answered to all the constructive comments of the reviewers.

Please find below our answers to reviewer 3 of our manuscript untitled “The p53 pathway and metabolism: the tree that hides the forest”.

Reviewer 3:

The review by Lahalle et al. focusses on the metabolic network that is controlled by the p53 pathway. The authors first summarize well studied aspects of p53’s functions in RNA repair, cell division, cell death and aging and then describe the components of a branch within the p53 pathway that pertains to metabolism. Subsequently, they elaborate on eight metabolic pathways controlled by p53 in detail.  

This review is interesting and well written. The figures are informative and I would assume that the proposed links between the p53 pathway and the control of metabolic networks will be inspiring for the field of cancer research.

Minor points:

I think that Figure 1 could be somewhat improved at the visual level (the proteins look like smarties, which I find a little distracting) and senescence should be spelled with a small s. Metabolism is linked to physiological features (muscle endurance), pathologies (obesity, tumorigenesis), a tissue (CNS), a state (stemness) and a biological process (senescence) in a seemingly random order. Could this be structured a bit more?

Since several reviewers found that some of the figures were slightly redundant, we removed Fig 1 and replaced it by a modified version of Fig 4. We believe this new Figure 1 illustrates more faithfully one of the main messages of this review regarding the complexity of the metabolic network regulated by the different components of the p53 pathway.

In my document there is no Figure 2.

This was unfortunately due to the software used by the journal which failed to incorporate this figure. This will be fixed during editing of the Ms.

Line 134: I suggest using the official gene symbol for Mieap (SPATA18) to avoid confusion.

Since several reports refer to MIEAP, we added both the official gene name SPATA18 and MIEAP to avoid any confusion.

Line 252: …profoundly affects pyrimidine but not purine biosynthesis.

The sentence has been changed.

Line 256: it seems that full gene names are written in italics occasionally (as compared to, for example, line 242). I guess this depends upon the Journal’s style but it should be harmonized throughout the text.

All genes are now indicated in italic.

Line 393: should it not read “for which the DNA binding properties are directly controlled by…”?

This sentence has been changed:

“p53 is a major effector of many metabolic stress responses by controlling the transcription of numerous metabolic genes [3]. Nevertheless, several studies have shown that its DNA binding properties are directly controlled by intracellular ROS, heme or ceramide levels, supporting the notion that p53 is also a bona fide metabolic sensor”.

Line 425: The sentence has a question mark at the end but it is not a question. If the authors intended it to be a question, they should invert “it” and “is”.

Indeed, there is no need for a question mark. We changed the sentence.